# Cost-effectiveness evaluation of different control strategies for *Clonorchis sinensis* infection in a high endemic area of China: A modelling study

Yun-Ting He[1]ᴑ, Xiao-Hong Huang[1,2]ᴑ, Yue-Yi Fang[3], Qing-Sheng Zeng[4], Lai-De Li[4], Le Luo[5], Ying-Si Lai[1,6]*

**1** Department of Medical Statistics, School of Public Health, Sun Yat-sen University, Guangzhou, Guangdong Province, People's Republic of China, **2** Statistics Office of network data information department, the First Affiliated Hospital of Guangzhou University of Traditional Chinese Medicine, Guangzhou, China, **3** Institute of Parasitic Diseases, Guangdong Provincial Center for Disease Control and Prevention, Guangzhou, Guangdong Province, People's Republic of China, **4** Xinhui District Center for Disease Control and Prevention, Jiangmen City, Guangdong Province, People's Republic of China, **5** Zhongshan Center for Disease Control and Prevention, Zhongshan, Guangdong Province, People's Republic of China, **6** Sun Yat-sen Global Health Institute, Sun Yat-sen University, Guangzhou, Guangdong Province, People's Republic of China

ᴑ These authors contributed equally to this work.
* laiys3@mail.sysu.edu.cn

**Editor:** jong-Yil Chai, Seoul National University College of Medicine, REPUBLIC OF KOREA

**Data Availability Statement:** Data used in the manuscript was derived from the published

## Abstract

Clonorchiasis is an important food-borne parasitic disease caused by *Clonorchis sinensis* infection. The evaluation of long-term cost-effectiveness of control strategies is important for disease control and prevention. The present study aimed to assess the cost-effectiveness of the three recommended strategies (i.e., WHO, Chinese and Guangdong strategies) and different combinations of commonly used measures (i.e., preventive chemotherapy, information, education, and communication (IEC) and environmental improvement) on clonorchiasis. The study area, Fusha town in Guangdong Province, was a typical high endemic area in China. The analysis was based on a multi-group transmission model of *C. sinensis* infection. We set the intervention duration for 10 years and post-intervention period for 50 years. The corresponding costs and DALYs were estimated. Strategies with incremental cost-effectiveness ratios (ICERs) less than 1/5 of the willingness-to-pay threshold were identified as highly cost-effective strategies. The optimal control strategy was obtained using the next best comparator method. The ICERs of Guangdong strategy were $172 (95% *CI*: $143-$230) US for praziquantel and $106 (95% *CI*: $85-$143) US for albendazole, suggesting the highest cost-effectiveness among the three recommended strategies. For praziquantel, 470 sets of control strategies were identified as highly cost-effective strategies for achieving infection control (prevalence<5%). The optimal strategy consisted of chemotherapy targeted on at-risk population, IEC and environmental improvement, with coverages all being 100%, and with the ICER of $202 (95% *CI*: $168-$271) US. The results for transmission control (prevalence<1%) and albendazole were obtained with the same procedures. The findings may help to develop control policies for *C. sinensis* infection in high

literature which has been included in the manuscript (Du S. et al [21]).

**Funding:** YSL received the financial support from the National Natural Science Foundation of China (project no. 82073665, http://www.nsfc.gov.cn/), the Natural Science Foundation of Guangdong Province (project no. 2022A1515010042, http://gdstc.gd.gov.cn/) and Sanming Project of Medicine in Shenzhen (grant no. SZSM201803061, http://wjw.sz.gov.cn/ztzl/smgc/). The funders had no role in study design, data collection and analysis, decision to publish, or preparation of the manuscript.

**Competing interests:** The authors have declared that no competing interests exist.

endemic areas. Moreover, the method adopted is applicable for assessment of optimal strategies in other endemic areas.

## Author summary

Clonorchiasis, a food-borne trematodiases, affects millions of people in Asia. Highly cost-effective control strategies are critical for its control. Previous studies considering the economic evaluation of control strategies were rare, mostly based on interventions in practical, and not capable of evaluating long-term cost-effectiveness of strategies with possible combinations of control measures or under various coverages. Based on a dynamic, multi-group transmission model, we simulated different control strategies in a high clonorchiasis endemic area, and evaluated their cost-effectiveness. Among the three recommended strategies (i.e., WHO, Chinese and Guangdong strategies), the Guangdong strategy was the most cost-effective. For praziquantel, 470 sets of control strategies were identified as highly cost-effective strategies for achieving infection control (prevalence<5%) among the strategies of possible combinations of the three common measures (i.e., preventive chemotherapy, information, education, and communication (IEC) and environmental modification). The optimal strategy consisted of chemotherapy targeted on at-risk population, IEC and environmental improvement, with coverages all being 100%. The results for transmission control (prevalence<1%) and albendazole were obtained with the same procedures. The numerical results may help to develop control strategies for *C. sinensis* infection in high endemic areas. The methodology is applicable for other different endemic areas.

## Introduction

Clonorchiasis, caused by *Clonorchis sinensis* infection, is one of the most important food-borne parasitic disease that affects more than 15 million people worldwide [1]. The global burden was estimated to be 522,863 disability-adjusted life years (DALYs) by WHO in 2010 [2]. Control strategies are vital in reducing *C. sinensis* infection, which commonly include preventive chemotherapy, information, education, and communication (IEC), environmental improvement, and possible combinations of the above measures [3]. Among them, chemotherapy is the major strategy for infection control, which can be either targeted on the whole population (i.e., mass drug administration, MDA) or selected population (e.g., individuals detected by stool examination, at-risk population with raw-fish-eating behavior) [3,4]. IEC aims to improve people's behaviors (i.e., developing hygienic habits and stopping raw-fish-consumption) to control *C. sinensis* infection. And environmental improvement can prevent faeces with *C. sinensis* eggs into water bodies by removing unimproved toilets built near fish ponds [5,6].

 While growing emphasis has been laid on implementing integrated control strategies to control *C. sinensis* infection, integrated control programs include combination of different measures, leading to increasing requirement of costs and labors. Therefore, besides the sustainability and long-term effects of control strategies, attention should also be paid to the cost-effectiveness, which could support local governments to better design appropriate control interventions. Previous studies considering the economic evaluation of control strategies were rare. Two studies performed cost-effectiveness analysis of *C. sinensis* control strategies [7,8],

however, they used prevalence to measure population health, rather than the one favored by the WHO-CHOICE, namely disability-adjusted life years (DALYs), which integrated morbidity and mortality of disease and has been widely used in cost-effectiveness analysis of parasitic diseases [9,10]. Moreover, most studies were based on intervention in practical [7,8,11,12], limited to the real interventions, and not capable of evaluating long-term cost-effectiveness of strategies with possible combinations of control measures or under various coverages. Mathematical transmission modelling, which was widely used in other neglected tropical diseases researches [13–15], has the capacity to simulate interventions with different parameters and enables important factors like chemotherapy compliance to be considered in the simulation, thus provides more specific information in implementing appropriate control strategies. However, for those models currently developed on *C. sinensis* infection, cost-effectiveness was not considered [16–20].

In this study, we aimed to conduct cost-effectiveness analysis on control strategies against *C. sinensis* infection based on a mathematical transmission model [20], with DALYs used as the effectiveness indicator. Control strategies being evaluated included the three recommended strategies (i.e., WHO strategy, Chinese government strategy and Guangdong Province government strategy) and possible combinations of common measures. We took Fusha Town, a typical high endemic area of *C. sinensis* infection (33.67% infection rate in 2017) [21], as the study area. It locates in Zhongshan City, Guangdong Province.

## Methods

### Methods overview

In this study, we modified the previous multi-group dynamic transmission model [20] to perform cost-effectiveness analysis of different control strategies over a 60-year period. Fusha town was taken as the study area, and the parameter estimation was done with Bayesian melding approach by fitting to the data of Fusha Town. Fusha town (area: 37 km$^2$, coordinates: 22° 40'2.64"N, 113°20'58.2"E, and population: 59,593 in 2017) is a typical high endemic area located in Zhongshan City, Guangdong Province, with flourishing fishing industry and raw fish-eating habits. The observed prevalence of *C. sinensis* infection of Fusha town was 33.67% in the year 2017 [21]. Detailed explanations of the updated transmission models can be found in S1 File.

To simulate interventions, a full model with interventions was modified based on the basic model, details of which can be found in S2 File. Simulated interventions included the three recommended strategies (i.e., WHO strategy, Chinese government strategy and Guangdong Province government strategy) and various combinations of three commonly used measures (i.e., chemotherapy, IEC and environmental improvement), coverage of which were set as different values to represent different combined strategies. Then, cost-effectiveness analysis was performed for each strategy to obtain the ones with high cost-effectiveness. The optimal cost-effective control strategy was further obtained by the next best comparator method [13].

### Simulation of interventions

We set the intervention duration for 10 years and post-intervention period for 50 years. The time horizon of intervention duration was determined based on the previous related study and the duration necessary to capture long-term differences between control strategies [22]. The first category of simulated interventions was the three control strategies recommended by WHO (chemotherapy with 100% coverage targeted on whole population), the Chinese government (chemotherapy targeted on at-risk population and environmental improvement, both with 80% coverage), and the government of Guangdong Province (chemotherapy with 80% coverage targeted on at-risk population, environmental improvement with 90% coverage and

IEC with 90% coverage). Detailed description and values of intervention parameters were displayed in S3 File.

The second category was the combinations of the three commonly used measures, intervention parameters of which were set to a series of values. Particularly, the coverage rate of chemotherapy ($C_m$), the proportion of people accepting IEC ($C_e$) and the coverage rate of environmental improvement ($C_d$) were set to different values (from 0% to 100%, with interval being 10%), where the targeted population of chemotherapy were either whole population, at-risk population with raw-fish-consumption behaviors or positive population, and the frequency of chemotherapy being once per year. Two types of drugs, namely praziquantel (PZQ) and albendazole (ABZ), were selected as chemotherapy options. A total of 3993 combinations were simulated. Strategies were recognized as sustainable ones with infection control or transmission control, if they were with control reproduction number ($Rc$)<1, reached infection control (with prevalence <5%) or transmission control (with prevalence <1%) status in 10 years, respectively, and continued to maintain in the post-intervention period (50 years) [20].

## Cost-effectiveness analysis

We performed cost-effectiveness analysis by estimating costs and DALYs of the corresponding control strategies, following procedures similar as other studies [13,14]. The components of total costs included chemotherapy, IEC and environmental improvement. Among them, the cost of chemotherapy included per capita drug cost and delivery cost (salaries, transportation, administrative fees, program running, community awareness activities, training and drug distribution, etc) [13,22]. If chemotherapy was targeted on positive population, diagnostic costs were extra included. The environmental improvement costs were estimated as initial investment costs and recurrent costs of toilets. The cost details and calculation formulas were explained in S4 File. All costs were transformed to 2020 US international dollars (I$) according to consumer price index (CPI) and purchasing power parities (PPP) in 2020 [23,24].

DALYs were used as the indicator of effectiveness, calculated as the sum of years of life lost (YLLs) and years lost due to disability (YLDs). YLLs were calculated as number of deaths from *C. sinensis* infection multiplying years of life lost for each death case. Assuming that the disability of individuals with *C. sinensis* is mainly from heavy infection [2,25,26], heavy infection cases were used to estimate YLDs. According to convention in assessing health outcomes from disease, we calculated YLDs with disability weight (DW) [27]. Detailed calculation method and value settings of the parameters were displayed in S5 File.

The cost-effectiveness of each strategy was shown as incremental cost-effectiveness ratio (ICER) compared to baseline scenario without any intervention, calculated as $\frac{C}{D_0 - D}$, where $C$ represents total costs. $D$ and $D_0$ represents DALYs with and without intervention, respectively. Strategies were identified as highly cost-effective strategies if the ICER was below a willingness-to-pay threshold, which was set to be the gross domestic product (GDP) per capita in 2020 (i.e., $17,307 US per DALY for China) [13,28]. Furthermore, we used the "next best comparator" approach to obtain the optimal control strategy when the targeted population of chemotherapy was set to different types (i.e., any type, whole, at-risk population or infected individuals) [13]. The first step was ranking all strategies by increasing averted DALYs, and ICER of each strategy was computed compared to the next most effective strategy (defined as the strategy with next highest averted DALYs). Expressed in formula is ICER $= \frac{C_1 - C_2}{D_2 - D_1}$. Here $C_1$ and $C_2$ represents the costs of the current strategy and its next most effective strategy, respectively, while $D_1$ and $D_2$ represents the DALYs of these two strategies. The optimal control strategy was defined as the one with the highest averted DALYs and the ICER below the willingness-to-pay threshold.

## Uncertainty analysis

We performed a multi-way uncertainty analysis by sampling 500 sets of parameters from the distributions of costs and DALYs-related parameters and combining them with 500 sets of transmission parameters drawn from their posterior distributions, which respectively reflected the uncertainty in cost-effectiveness and that in disease transmission. In each simulation based on a set of parameters, ICER was computed through next best comparator method and the optimal strategy was obtained. The overall optimal cost-effective strategy after considering uncertainty was defined as the one with the highest proportion among the 500 simulations.

Since the 100% coverage rate of interventions might be difficult to realize in practice, we also estimated the optimal strategies by removing the effective control strategies with the coverage rate of any single measure being 100% out of the feasibility consideration, thus to obtain the optimal strategies with the upper limit of coverage rate being 90%. All the numerical computations were performed in MATLAB R2021b.

## Results

### The three recommended strategies

The costs and averted DALYs of the three recommended strategies for *C. sinensis* infection were shown in Table 1. We estimated that under the strategies recommended by the government of Guangdong Province, the highest DALYs were averted, reducing 96.6% (95% *CI*: 95.7%-97.9%) of the total DALYs under baseline scenario. When the drug was PZQ, the ICERs of the three recommended strategies compared to baseline scenario were $510 (95% *CI*: $394-$648) US, $136 (95% *CI*: $114-$180) US and $172 (95% *CI*: $143-$230) US, respectively. When the drug was ABZ, the ICERs were lower than that when the drug was PZQ. With "next best comparator" approach, Guangdong strategy averted the highest DALYs and the ICER compared to next most effective strategy (i.e., Chinese strategy) was within willing to pay threshold for both PZQ and ABZ, suggesting that Guangdong strategy was the optimal among the three recommended strategies. In contrast, the strategy recommended by the WHO averted the lowest DALYs and had the highest costs, suggesting lowest cost-effectiveness.

### Highly cost-effective control strategies

Totally, 470 sets and 42 sets of control strategies satisfied the condition for infection control and transmission control, respectively. When the drug was PZQ or ABZ, among the control strategies that could reach infection control or transmission control, all sets of strategies were with ICERs less than 1/5 of the willingness-to-pay threshold, suggesting that they were all highly cost-effective. Under the simulations with the best set of parameters, the total costs and

**Table 1. The costs and averted DALYs of the three recommended strategies for *C. sinensis* infection\*.**

| Indicators | WHO | The Chinese government | The government of Guangdong Province |
|---|---|---|---|
| Averted DALYs | 9,830.3 (7,719.7–12,722.4) | 12,155.7 (9,274.1–14,755.0) | 13,087.5 (9,870.7–15,899.7) |
| Reduction of DALYs under baseline scenario (%) | 72.5% (68.5%-85.1%) | 89.7% (88.5%-92.9%) | 96.6% (95.7%-97.9%) |
| Total costs (drug being PZQ) | 5,016,313 (4,959,156–5,068,060) | 1,649,335 (1,569,281–1,774,516) | 2,255,616 (2,124,141–2,434,023) |
| Total costs (drug being ABZ) | 1,686,324 (1,626,972–1,737,243) | 780,038 (730,479–847,596) | 1,386,281 (1,262,275–1,522,161) |
| Costs per DALY averted (PZQ) | 510 (394–648) | 136 (114–180) | 172 (143–230) |
| Costs per DALY averted (ABZ) | 172 (132–218) | 64 (54–85) | 106 (85–143) |

\*Data was presented as median (95% *CI*).

averted DALYs of the sustainable control strategies were shown in Appendix (S7 and S8 Tables).

## The optimal control strategies

**Infection control.** When chemotherapy was targeted on any population category (i.e., either whole, at-risk or positive population), for PZQ, the chemotherapy targeted on at-risk population combined with IEC and environmental improvement with coverage all being 100% was the optimal strategy to reach infection control in 48.0% of simulations. For drug ABZ, the chemotherapy targeted on whole population combined with IEC and environmental improvement with coverage all being 100% was the optimal strategy to reach infection control in 84.6% of simulations (Table 2). Using PZQ and ABZ as drugs, these two strategies were regarded as overall optimal control strategies with the highest proportions in simulations, respectively, and the corresponding ICERs were $202 (95% CI: $168-$271) US and $205 (95% CI: $166-$272) US, respectively. If we set the upper limit of coverage rate as 90%, the overall optimal strategies for PZQ and ABZ were still the combination of three measures, with the coverage rates all being the highest 90% (S9 Table), but the targeted population of chemotherapy changed to positive population for PZQ.

## Transmission control

When chemotherapy was targeted on any population category, for drug PZQ, the chemotherapy targeted on positive population combined with IEC and environmental improvement with coverage all being 100% was optimal to reach transmission control in 31.6% of simulations, and the ICER was $339 (95% CI: $265-$453) US. For drug ABZ, the overall optimal strategy was the same as that for infection control, namely the chemotherapy targeted on whole population combined with IEC and environmental improvement with coverage all being 100%, which was optimal in 97.2% of simulations (Table 2). If we set the upper limit of coverage rate as 90%, the optimal strategies with the largest proportion were the same as that for infection control (S9 Table). Due to the low prevalence requirement of transmission control, there was no strategy with chemotherapy targeted on at-risk population met the required effectiveness and sustainability. The optimal strategies for other targeted population of chemotherapy were displayed in Appendix (S10 and S11 Tables).

**Table 2. The optimal cost-effective strategies for any targeted population of chemotherapy\*.**

| Control type | Drug | Targeted population of chemotherapy | The optimal strategy | | | |
|---|---|---|---|---|---|---|
| | | | $C_d$ | $C_e$ | $C_m$ | Proportion (%) |
| Infection control | PZQ | At-risk | 1.00 | 1.00 | 1.00 | 240 (48.0) |
| | | Positive | 1.00 | 1.00 | 1.00 | 129 (25.8) |
| | ABZ | Whole | 1.00 | 1.00 | 1.00 | 423 (84.6) |
| | | At-risk | 1.00 | 1.00 | 1.00 | 66 (13.2) |
| Transmission control | PZQ | Positive | 1.00 | 1.00 | 1.00 | 158 (31.6) |
| | | Positive | 1.00 | 1.00 | 0.80 | 90 (18.0) |
| | | Positive | 1.00 | 1.00 | 0.70 | 88 (17.6) |
| | | Positive | 1.00 | 1.00 | 0.60 | 64 (12.8) |
| | | Positive | 1.00 | 1.00 | 0.90 | 50 (10.0) |
| | ABZ | Whole | 1.00 | 1.00 | 1.00 | 486 (97.2) |

\*Only strategies with proportions≥10% were presented. The frequency of chemotherapy was once a year, and the intervention duration was 10 years. Targeted population of chemotherapy being any means that it could be any type (i.e., whole, at-risk or positive).

## Discussion

In this study, we modified the previous multi-group dynamic transmission models to evaluate the cost-effectiveness of different control strategies on *C. sinensis* infection under a typical high endemic environment. Among the three recommended strategies, the strategy of Guangdong Province, which was with a combination of three common measures, had the highest cost-effectiveness. For drug PZQ, chemotherapy targeted on at-risk population or on positive population combined with IEC and environmental improvement with coverages all being 100% were the optimal strategies among all possible combinations to achieve infection control or transmission control, respectively. For drug ABZ, chemotherapy targeted on whole population combined with IEC and environmental improvement with coverages all being 100% were the optimal strategies to achieve infection control or transmission control.

### The optimal control strategies

The results suggested that optimal cost-effective strategies tended to be strategies that combined three common measures (i.e., chemotherapy, IEC and environmental improvement), which was consistent with the conclusions of previous studies that implementation of integrated strategies based on multiple interventions could be successful to control parasitic diseases [29]. Because PZQ was relatively expensive in China market, chemotherapy targeted on whole population could lead to much higher costs and did not avert much more DALYs compared to that targeted on at-risk or positive populations, thus was not part of the optimal long-term cost-effective strategy for either infection or transmission control. As suggested by previous study, chemotherapy targeted on at-risk population had potential to be the best [12]. However, it was not able to realize transmission control. Chemotherapy targeted on positive population was suggested to be effective in reducing infection rate, so that it might be highly cost-effective for achieving control status with strict requirements [30]. When the drug was ABZ, a lower price drug compared to PZQ, the optimal strategies were targeted on whole population. Possible reasons might be that when chemotherapy was targeted on positive individuals, extra diagnostic costs were needed for population mass screening, and conventional microscopic techniques such as the Kato-Katz diagnostic approach were considered insufficiently sensitive, leading to missed diagnosis and lack of subsequent chemotherapy treatment for some positive people [31]. In this case, chemotherapy targeted on whole population did not cost much due to the cheaper cost of the drug.

The optimal strategies for both infection control and transmission control required the high coverage (i.e., 100%) of the three measures, which was necessary in elimination of parasitic diseases [32]. Considering that 100% coverage may not be easy to achieve in practice, we set the upper limit of coverage as 90% and performed the analysis. In this case, when the drug was ABZ, the optimal strategies for both infection control and transmission control were still the combination of three measures with the highest coverage (i.e., 90%), and the chemotherapy was targeted on whole population. While for the drug PZQ, the optimal strategies were with the targeted population of chemotherapy being positive individuals. The possible reason was that for drug PZQ, a 10% reduction in chemotherapy coverage reduced the overall costs of drugs and diagnosis and made the strategy based on chemotherapy targeted on positive population more cost-effective than the one based on at-risk population. However, as the cost of ABZ was much lower than that of PZQ, a 10% reduction in coverage did not lead to much decrease of overall costs due to the low costs of drugs, so that the optimal strategy was still based on chemotherapy targeted on whole population.

## Implications for policy making

In uncertainty analysis, all optimal strategies with high proportions ($\geq$10%) in simulations were the combined ones. Moreover, they included IEC and environmental improvement with 100% coverages, suggesting that the two measures played significant roles to achieve a sustainable and cost-effective control of *C. sinensis* infection, which should be fully considered in control programs. It was consistent with previous studies that the combination of health education and chemotherapy was more effective than chemotherapy alone for controlling infection control [33,34]. Considering that IEC could also promote compliance of chemotherapy and bring economic benefits with its low price, governments may lay emphasis on training health committees to deliver IEC [34]. Particularly, for children at school-age, health education is more likely to increase knowledge on preventing *C. sinensis* infection and promoting belief of stopping eating raw fish [35].

For environmental improvement, residents in less developed areas may lack awareness of constructing sanitation toilets, and the established sanitation toilets may lack management after completion of construction, which bring difficulties in increasing the coverage [36]. In this case, the government should raise public awareness of the importance of constructing sanitation toilets, and provide subsidies to increase public motivation [37]. Besides, attentions should be paid to management and maintenance.

Previous studies suggested that PZQ and ABZ had similar efficacy, so that we considered both drugs as mass chemotherapy options to make evaluation of control strategies more comprehensive [38]. Our results suggested that the strategies with the chemotherapy targeted on positive population were optimal for PZQ, namely the more expensive drug, to achieve transmission control. Although disadvantages existed for the current diagnostic methods (e.g., low profitability, high technical requirements, low sensitivity) [3,39], researchers suggested that the compliance of chemotherapy in the positive population was higher, which was feasible for chemotherapy in long-term interventions [40]. We further set the ICER equal to the willingness-to-pay threshold to estimate the maximum acceptable diagnostic cost. Even if the diagnostic cost increased ten times, the strategy targeted on positive population was still cost-effective. Studies have suggested that chemotherapy targeted on positive population could be used as control strategies in low endemic areas [30]. Moreover, compared to massive chemotherapy, it might have lower cost per DALY averted [12]. In case that the compliance of chemotherapy was very low in whole or at-risk population, or the drug price was very high, chemotherapy targeted on positive individuals might be an alternative to massive or selective chemotherapy. However, developing more sensitive diagnostic approaches need to be emphasized. Regarding the two drugs, PZQ and ABZ had their own features. PZQ was widely used in practice and has a short medication time, but with poor taste, peculiar smell, and some side effects. ABZ was odorless and tasteless, and with good security in medication [41], so that we regarded ABZ as a possibility of chemotherapy. Nevertheless, the drug type chosen should follow local conditions, and the corresponding optimal cost-effective strategy may be different.

## Comparison with previous studies

Only a few studies have investigated the cost-effectiveness of control strategies on *C. sinensis* infection, and they were all conducted from the perspective of real intervention. Two studies proposed recommended strategies with high cost-effectiveness. Fang and colleagues suggested that chemotherapy targeted on active medication population combined with IEC had the lowest cost per 1% reduction in infection rates, compared with chemotherapy targeted on positive population, or chemotherapy targeted on whole population [8]. Qian et al's study showed that selective chemotherapy targeted on at-risk population was the best among three strategies (i.e.,

massive, selective and individual chemotherapy) [12]. The recommended strategies of the two studies were different from our study, but restricted to some limitations. Firstly, they only considered the effect and cost-effectiveness during the intervention period and ignored the long-time influence of the strategies that may lead to future epidemic rebound. Due to the perspective of real interventions, sensitivity of diagnosis was not considered, thus some positive individuals may be missed. Furthermore, they were not capable of evaluating cost-effectiveness of strategies with different coverages and different combinations to provide detailed numerical simulation results. Only chemotherapy and IEC were evaluated as control measures, environmental improvement was not included.

## Strengths and limitations

We performed the cost-effectiveness analysis based on a mathematical transmission model and used DALYs as indicators, which have not been investigated in previous studies on *C. sinensis* infection. The transmission model took into account the recurrence of infection and was able to further evaluate the long-term cost-effectiveness of specific strategies. Compared to studies conducted from real intervention perspective, the analysis in our study was not restricted to limited number of interventions and could evaluate strategies with different coverages and combinations, as various intervention parameters were set in the model, including efficacy and compliance parameters. Some factors such as diagnosis costs may be different due to regional differences, and the cost-effectiveness analysis could take the uncertainty of cost into account by setting ranges of corresponding parameters. Furthermore, the uncertainty analysis performed in our study provided more information than just the point estimation.

There are some limitations in this study. Regarding the characteristics of the transmission model, several potential influencing factors were not considered, including the influence of area interaction, reservoir hosts and seasons, which were specifically discussed in our previous study [20]. Moreover, due to the difficulty in obtaining corresponding data, some heterogeneity issues, such as compliance and efficacy among different population groups, were not considered in the model [20]. For DALY calculation, mild and moderate infections might also cause the loss of healthy life years [42]. However, due to the lack of disability weight information for mild and moderate infections and the weights might be very small, we did not consider the loss of healthy life years resulted from these infections, similar as most previous studies on estimation of the disease burden of clonorchiasis [2,43]. The transmission model was applied in a specific highly endemic area (infection rates ≥ 20%) with flourishing fishing industry and raw fish-eating habits. The results of the optimal strategies derived from this study may not be applicable to other moderate or low endemic areas. However, the numerical simulation results can still provide important information for the selection of control strategies, and the method adopted in this study is also applicable for the selection of the optimal strategies in other areas.

## Conclusion

In this study, the long-term cost-effectiveness of control strategies for *C. sinensis* infection was evaluated with different coverages and combinations of control measures in a highly endemic area of China. The findings may help to formulate disease prevention and control policies for *C. sinensis* infection in highly endemic areas. Moreover, the method adopted in this study is also applicable for the selection of the optimal strategies in other endemic regions.

## Supporting information

**S1 Fig. The full model of *C. sinensis* infection with interventions.**
(TIF)

**S1 Table. Descriptions of model parameters (unit: day$^{-1}$).**
(DOCX)

**S2 Table. Observed prevalence of *C. sinensis* infection among groups of people with different frequencies of raw fish consumption in Fusha Town.**
(DOCX)

**S3 Table. The prior distributions and posterior estimations of model parameters (unit: day$^{-1}$).**
(DOCX)

**S4 Table. Values set for compliance of chemotherapy among different kinds of targeted population.**
(DOCX)

**S5 Table. Values of intervention parameters of the current recommended control strategies.**
(DOCX)

**S6 Table. Costs of each aspect of the control strategies.**
(DOCX)

**S7 Table. Simulation results under the effective control strategies that could reach infection control within 10 years.**
(DOCX)

**S8 Table. Simulation results under the effective control strategies that could reach transmission control within 10 years.**
(DOCX)

**S9 Table. The optimal cost-effective strategies with targeted population of any types and the upper coverage set to 90%.**
(DOCX)

**S10 Table. The optimal cost-effective strategies to reach infection control for other targeted populations of chemotherapy.**
(DOCX)

**S11 Table. The optimal cost-effective strategies to reach transmission control for other targeted populations of chemotherapy.**
(DOCX)

**S1 File. Transmission model.**
(DOCX)

**S2 File. Full model with interventions.**
(DOCX)

**S3 File. Three recommended strategies.**
(DOCX)

**S4 File. Calculation of costs.**
(DOCX)

**S5 File. Calculation of DALYs.**
(DOCX)

## Author Contributions

**Conceptualization:** Yue-Yi Fang, Qing-Sheng Zeng, Lai-De Li, Le Luo, Ying-Si Lai.

**Data curation:** Yun-Ting He, Xiao-Hong Huang.

**Formal analysis:** Yun-Ting He, Xiao-Hong Huang, Ying-Si Lai.

**Funding acquisition:** Ying-Si Lai.

**Investigation:** Yun-Ting He, Xiao-Hong Huang, Ying-Si Lai.

**Methodology:** Yun-Ting He, Xiao-Hong Huang, Ying-Si Lai.

**Project administration:** Yun-Ting He, Xiao-Hong Huang.

**Resources:** Ying-Si Lai.

**Software:** Yun-Ting He, Xiao-Hong Huang, Ying-Si Lai.

**Supervision:** Ying-Si Lai.

**Validation:** Yun-Ting He, Xiao-Hong Huang, Ying-Si Lai.

**Visualization:** Yun-Ting He, Xiao-Hong Huang, Ying-Si Lai.

**Writing – original draft:** Yun-Ting He, Xiao-Hong Huang, Ying-Si Lai.

**Writing – review & editing:** Yue-Yi Fang, Qing-Sheng Zeng, Lai-De Li, Le Luo, Ying-Si Lai.

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
