## [Decision Letter · Decision Letter 0]

24 Jan 2022

Dear Dr. Lai,

Thank you very much for submitting your manuscript "Assessment of different control strategies against Clonorchis sinensis infection in high endemic areas: a cost-effectiveness modelling study" for consideration at PLOS Neglected Tropical Diseases. As with all papers reviewed by the journal, your manuscript was reviewed by members of the editorial board and by several independent reviewers. In light of the reviews (below this email), we would like to invite the resubmission of a significantly-revised version that takes into account the reviewers' comments. 

Your manuscript has been reviewed by two independent reviewers. They recommended major revision to your manuscript. Please carefully read the Reviewer 2's comments and address the points if you agree to revise your manuscript.

We cannot make any decision about publication until we have seen the revised manuscript and your response to the reviewers' comments. Your revised manuscript is also likely to be sent to reviewers for further evaluation.

Sincerely,

jong-Yil Chai

Associate Editor

Aaron Jex

Deputy Editor

Your manuscript has been reviewed by two independent reviewers. They recommended major revision to your manuscript. Please carefully read the Reviewer 2's comments and address the points if you agree to revise your manuscript.

Reviewer's Responses to Questions

**Key Review Criteria Required for Acceptance?**

**Methods**

-Are the objectives of the study clearly articulated with a clear testable hypothesis stated?

-Is the study design appropriate to address the stated objectives?

-Is the population clearly described and appropriate for the hypothesis being tested?

-Is the sample size sufficient to ensure adequate power to address the hypothesis being tested?

-Were correct statistical analysis used to support conclusions?

-Are there concerns about ethical or regulatory requirements being met?

Reviewer #1: (No Response)

Reviewer #2: ① Please describe the materials and methods more clearly and systematically. 

② Please replace the previously well-known methods as references.

③ Please define the population characters of a subjected endemic area, Fusha Town.

④ Please refer the guidelines of PLOS NTD.

**Results**

-Does the analysis presented match the analysis plan?

-Are the results clearly and completely presented?

-Are the figures (Tables, Images) of sufficient quality for clarity?

Reviewer #1: (No Response)

Reviewer #2: ① Please describe the results more clearly and systematically. 

② Please refer the guidelines of PLOS NTD.

**Conclusions**

-Are the conclusions supported by the data presented?

-Are the limitations of analysis clearly described?

-Do the authors discuss how these data can be helpful to advance our understanding of the topic under study?

-Is public health relevance addressed?

Reviewer #1: (No Response)

Reviewer #2: Please describe the conclusion more objectively based on the findings obtained by the present study. 

-> In this study, the long term cost-effectiveness for control strategies of clonorchiasis was evaluated with different coverages and combinations of control measures in a highly endemic area of China. The findings may be help to make disease prevention and control policies for clonorchiasis in highly endemic areas

**Editorial and Data Presentation Modifications?**

Reviewer #1: (No Response)

Reviewer #2: (No Response)

**Summary and General Comments**

Reviewer #1: (No Response)

Reviewer #2: This manuscript is on the the cost-effectiveness evaluation about three different control strategies for clonorchiasis in a highly endemic area of China. It has important informations on the cost-effectiveness in control strategies for clonorchiasis. However, it has lots of limited points to be published in PLOS NTD. First of all, this manuscript should be revised English by a native speaker. It is highly questionable whether albendazole is an effective drug for clonorchiasis chemotherapy.

1. Title: Please refer the revised title. 

“Evaluation of cost-effectiveness about three different control strategies for Clonorchis sinensis infection in a highly endemic area of China”or

“Cost-effectiveness evaluation about three different control strategies for clonorchiasis in a highly endemic areas of China

2. Abstract 

① Please briefly describe the purpose of this study in the beginning. 

Line 21-28: Clonorchiasis is an important food-borne parasitic disease caused by Clonorchis sinensis infection. Long-term cost-effectiveness of control strategies needs to be evaluated to provide information for establishing control programs. We performed cost effectiveness analysis of three recommended strategies (i.e., WHO, Chinese and Guangdong strategies) and different combinations of measures (i.e., preventive 26 chemotherapy, information, education, and communication (IEC) and environmental modification) under a high endemic environment, based on a multi-group transmission model of C. sinensis infection.

-> Present study was performed to evaluate the cost effectiveness on 3 different control strategies, i.e., WHO, Chinese and Guangdong, for clonorchiasis and also analized the effects of different control measures, i.e., preventive chemotherapy, information, education, and communication (IEC) and environmental renovation, in a highly endemic region in China based on a multi-group transmission model of C. sinensis infection. 

② Please describe the materials and methods more clearly and briefly. 

Line 28-30. We set the intervention duration for 10 years and post intervention period for 50 years. The optimal control strategy was obtained using next best comparator method. Uncertainty analysis was further conducted. + informations on the subjected region and disability-adjusted life years (DALYs).

③ Please describe the results more clearly and systematically. 

Line 30-42: We found that Guangdong strategy was most cost-effective among the three recommended strategies, with cost per DALY averted being $172 (95% CI: $143-$230) US for praziquantel and $106 (95% CI: $85-$143) US for albendazole. The optimal strategies with high proportions in uncertainty analysis tended to be the ones combined with preventive chemotherapy, IEC and environmental modification, under coverages of all being 100%. For praziquantel, chemotherapy of the optimal strategies for infection control (maintaining a prevalence<5%) and transmission control (maintaining a prevalence<1%) were targeted on at-risk population and positive population, respectively, with coverages 100% and costs per DALY averted $202 (95% CI: $168-40 $271) US and $339 (95% CI: $265-$453) US, respectively. For albendazole, the chemotherapy of optimal strategies was targeted on whole population, and the costs per DALY averted was $205 (95% CI: $166-$272) US. 

-> In Guangdong strategy, the cost per DALY (disability-adjusted life year) avrted were $172 (95% CI: $143-$230) US for praziquantel and $106 (95% CI: $85-$143) US for albendazole, which was most cost-effective among 3 control strategies for clonorchiasis. 

Please reveal the evidences (findings) as numerical results you obtained and don’t make them descriptively.

④ Please describe the conclusion more objectively based on the findings obtained by the present study. The conclusive remarks are too long.

Line 42-47: In this study, we evaluated the long term cost-effectiveness of control strategies with different coverages and combinations, not restricted to limited intervention types. The numerical results provided information for developing disease prevention and control policies on C. sinensis infection in high endemic areas. Moreover, the method adopted in this study is applicable for assessment of optimal strategies in other endemic areas.

-> In this study, the long term cost-effectiveness for control strategies of clonorchiasis was evaluated with different coverages and combinations of control measures in a highly endemic area of China. The findings may be help to make disease prevention and control policies for clonorchiasis in highly endemic areas. 

3. Introduction 

① Please describe the background of this study more specifically. 

Line 50: Caused by infection with Clonorchis sinensis, clonorchiasis is one of the most

-> Clonorchiasis, Clonorchis sinensis infection, is one of the 

Line 55: environmental modification -> environmental renovation

Line 60: people’s healthy behaviors -> people’s behaviors

hygiene habits -> hygienic habits

② Please describe the purpose elicited from backgrounds of this study in the last paragraph. 

Line 91: Please more detailedly designate “Fusha Town”(location and/or province in China) and reveal the references for the clue of clonorchiasis endemic.

PLOS authors have the option to publish the peer review history of their article (what does this mean?). If published, this will include your full peer review and any attached files.

Reviewer #1: No

Reviewer #2: No
---

## [Decision Letter · Decision Letter 1]

18 Apr 2022

Dear Dr. Lai,

We are pleased to inform you that your manuscript 'Cost-effectiveness evaluation of different control strategies for Clonorchis sinensis infection in a high endemic area of China: a modelling study' has been provisionally accepted for publication in PLOS Neglected Tropical Diseases.

Best regards,

jong-Yil Chai

Associate Editor

Aaron Jex

Deputy Editor

This revised manuscript is on the cost-effectiveness evaluation about 3 different control strategies for clonorchiasis in a highly endemic area of China. It is much improved by the revision to be published in PLOS NTD by the revision.

Reviewer's Responses to Questions

**Key Review Criteria Required for Acceptance?**

**Methods**

-Are the objectives of the study clearly articulated with a clear testable hypothesis stated?

-Is the study design appropriate to address the stated objectives?

-Is the population clearly described and appropriate for the hypothesis being tested?

-Is the sample size sufficient to ensure adequate power to address the hypothesis being tested?

-Were correct statistical analysis used to support conclusions?

-Are there concerns about ethical or regulatory requirements being met?

Reviewer #1: (No Response)

**Results**

-Does the analysis presented match the analysis plan?

-Are the results clearly and completely presented?

-Are the figures (Tables, Images) of sufficient quality for clarity?

Reviewer #1: (No Response)

**Conclusions**

-Are the conclusions supported by the data presented?

-Are the limitations of analysis clearly described?

-Do the authors discuss how these data can be helpful to advance our understanding of the topic under study?

-Is public health relevance addressed?

Reviewer #1: (No Response)

**Editorial and Data Presentation Modifications?**

Reviewer #1: (No Response)

**Summary and General Comments**

Reviewer #1: (No Response)

PLOS authors have the option to publish the peer review history of their article (what does this mean?). If published, this will include your full peer review and any attached files.

Reviewer #1: No

---

## [Editor Report · Acceptance letter]

19 May 2022

Dear Dr. Lai,

We are delighted to inform you that your manuscript, "Cost-effectiveness evaluation of different control strategies for Clonorchis sinensis infection in a high endemic area of China: a modelling study," has been formally accepted for publication in PLOS Neglected Tropical Diseases.

Best regards,

Shaden Kamhawi

co-Editor-in-Chief

Paul Brindley

co-Editor-in-Chief
